# A Simple Scaling Model for Bootstrapped DQN

## Abstract

We present a large-scale empirical study of Bootstrapped DQN (BDQN) and Randomized-Prior BDQN (RP-BDQN) in the DeepSea environment designed to isolate and parameterize exploration difficulty. Our primary contribution is a simple scaling model that accurately captures the probability of reward discovery as a function of task hardness and ensemble size. This model is parameterized by a method-dependent effectiveness factor, $\psi$. Under this framework, RP-BDQN demonstrates substantially higher effectiveness ($\psi \approx 0.87$) compared to BDQN ($\psi \approx 0.80$), enabling it to solve more challenging tasks. Our analysis reveals that this advantage stems from RP-BDQN's sustained ensemble diversity, which mitigates the posterior collapse observed in BDQN. Furthermore, we show how systematic deviations from this simple model diagnose complex second-order dynamics: we mechanistically link "cooperative" deviations in small ensembles to information sharing via the replay buffer, and "saturation" in large ensembles to correlated failures. We conclude by translating these findings into a prescriptive framework, offering practical guidance for configuring ensembles in deep exploration.

## 1 Introduction

In many areas of machine learning, algorithms learn from what are known as 'dense' rewards. This typically means that for every input the algorithm processes, there is a clearly defined desired output, or a straightforward way to calculate a 'goodness' score for the algorithm's prediction. Such a setup simplifies learning process, allowing for the direct application of backpropagation-based algorithms to adjust the model parameters. In contrast, the objective of Reinforcement Learning (RL) algorithms is to maximize the total reward accumulated over a sequence of interactions with an environment (Montague, 1999). Unlike typical supervised learning, successful RL algorithms must address two fundamental challenges stemming from this interaction model (Osband et al., 2020):

- **Exploration-exploitation tradeoff**: The agent needs to decide whether to exploit actions known to be effective or to explore new, potentially better (or worse) actions. This involves strategically prioritizing which feedback to learn from.

- **Long-term consequences**: The agent must consider how its current actions might affect opportunities and outcomes far into the future, beyond the immediate next step.

One approach to tackle the exploration-exploitation tradeoff is posterior sampling, where an agent maintains a distribution over optimal policies and acts according to samples from this belief. While effective in simpler settings, maintaining an exact posterior is computationally intractable for high-dimensional, non-linear function approximators, such as deep neural networks. This has driven the development of approximate methods that can capture the benefits of posterior sampling without the prohibitive computational cost.

In this work, we investigate the behavior of bootstrap-based posterior sampling for exploration. Bootstrapped DQN (BDQN) (Osband et al., 2016a) is a prominent example, training an ensemble of Q-value functions on different subsets of experience. While empirically successful, BDQN is known to suffer from posterior collapse, where all ensemble members converge to a single, overconfident estimate, thereby losing the diversity

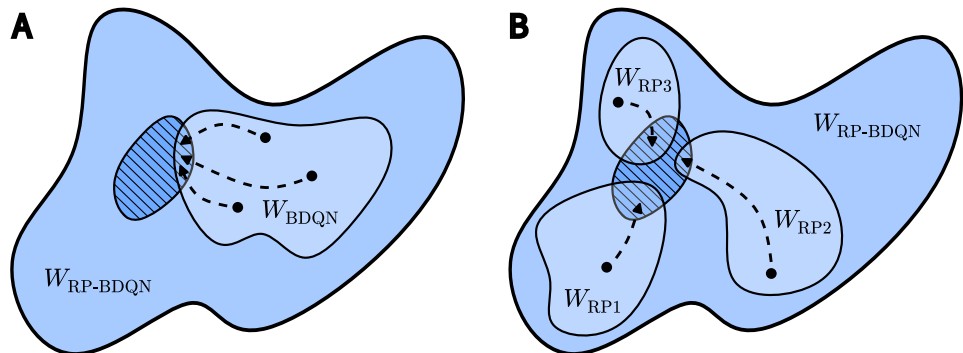

Figure 1: Schematic parameter-space view of how bootstrap ensembles form surrogate posteriors.
**A** - Bootstrapped DQN with ensemble size $K = 3$. All ensemble members are attracted toward the *same* low-loss basin (hatched region in $W_{\mathrm{BDQN}}$). The training dynamics (black arrows) funnel all members into the single low-loss region, causing the surrogate posterior to become sharply concentrated and epistemic uncertainty to collapse.
**B** - Bootstrapped DQN with Randomized Priors, ensemble size $K = 3$. Each ensemble member $W_{\mathrm{RP1}}$, $W_{\mathrm{RP2}}$, $W_{\mathrm{RP3}}$ carries an independent, frozen prior network that shifts its loss surface, producing distinct low-loss regions. The diversity is naturally preserved, and the posterior collapse is structurally prevented by the hard constraint on the parameter space.

needed for sustained exploration. As a remedy, Randomized-Prior BDQN (RP-BDQN) (Osband et al., 2018) was introduced, which adds a unique, frozen prior network to each ensemble member. This modification is designed to anchor each member in a different part of the parameter space, thereby enforcing diversity and improving exploration, as illustrated in Figure 1.

**Contributions**

- We present a systematic analysis of BDQN and RP-BDQN across over 40,000 configurations, revealing that a simple scaling model can characterize the convergence behavior of both methods. We show that despite ignoring ensemble interactions, this model explains a large fraction of the performance variance, suggesting that a 'best-of-K' independent trial viewpoint is a powerful first-order approximation.

- We provide evidence that RP-BDQN extends the performance boundary beyond where BDQN breaks down and exhibits consistently more robust convergence, linking this to its diversity-preserving mechanism.

- We offer a diagnostic framework using a controlled environment and a simple baseline model to facilitate future studies on epistemic exploration. We demonstrate that analyzing where such simple models fail is as informative as where they succeed, revealing mechanisms of implicit cooperation and correlated failure.

Taken together, our results offer an easy-to-follow guide for practitioners: RP-BDQN consistently outperforms BDQN, and the effect of using ensemble sizes larger than 10 diminishes significantly. For theorists, our findings raise an intriguing question: Why does a model that ignores inter-agent dynamics still capture the bulk of the performance, and what do the model's systematic failures tell us about the nature of these dynamics?

## 2 Background

**Markov Decision Process.** We model the environment as a Markov Decision Process (MDP), where an agent learns an optimal policy $\pi^\star$ to maximize discounted cumulative rewards. Because the environment's transition dynamics $P(s' \mid s, a)$ are unknown, the agent cannot plan and must learn through interaction. This paper focuses on epistemic exploration: systematically reducing the agent's uncertainty about the environment to discover high-reward behaviors efficiently.

**Deep Q-Learning.** Q-learning algorithms learn the optimal action-value function, $Q^\star(s, a)$, by iteratively updating an estimate based on the Bellman equation. While tabular representations are feasible for small state spaces, modern applications require non-linear function approximation. Deep Q-Networks (DQN) (Mnih et al., 2013) employ a neural network $Q_\theta(s, a)$ for this purpose. The network is trained with stochastic gradient descent to minimize the temporal-difference error:

$$\mathcal{L}(\theta) = \mathbb{E}_{(s,a,r,s') \sim \mathcal{D}}\Big[\big(r + \gamma \max_{a'} Q_{\theta^-}(s', a') - Q_\theta(s, a)\big)^2\Big]$$

where training is stabilized by using a replay buffer $\mathcal{D}$ and a periodically updated target network with parameters $\theta^-$.

**Posterior sampling for exploration.** In simpler, small-scale, tabular MDPs, where all states and actions can be explicitly enumerated, the exploration-exploitation tradeoff is well-understood and can be managed using principled methods like posterior sampling (Osband et al., 2016b; Strens, 2000). During posterior sampling, the agent maintains a posterior distribution (belief) over plausible value functions (or models), and acts according to samples drawn from this distribution. However, for the complex, non-linear function approximations (typically deep neural networks) that are central to modern deep reinforcement learning, performing exact posterior inference is computationally prohibitive. This creates a need for tractable surrogates: practical methods that can (i) approximate Bayesian-like exploration closely enough to guide the agent effectively and (ii) scale to models with potentially millions of parameters.

Several approaches aim to approximate the Bayesian posterior to achieve efficient exploration. Some methods aim for direct approximation, for instance, using Markov Chain Monte Carlo techniques (Hastings, 1970; Ishfaq et al., 2023), Sequential Monte Carlo (Van der Vaart et al., 2024) or variational approximations to the Bayesian posterior (Srivastava et al., 2014; Fortunato et al., 2018). In contrast, an alternative line of research, and a focus of this work, utilizes the stationary distribution of the optimization algorithm itself as a surrogate posterior (Osband et al., 2016a; 2018). While it has been shown that stochastic gradient descent iterates can yield a posterior that is approximately Bayesian under certain conditions (Mandt et al., 2017), the properties of stationary distributions for more complex optimizers, such as Adam (Kingma & Ba, 2015), are less characterized. Nevertheless, a significant benefit of using such surrogate posteriors is the simplicity and efficiency of sampling from them, making them an attractive choice for driving exploration.

**Bootstrap Ensembles as Approximate Posterior Distributions.** Bootstrap sampling mimics Thompson sampling (Thompson, 1933) by maintaining $K$ different value function estimates, each trained on a stochastically resampled subset of data. In the linear setting (where value functions are linear models), Randomized Least-Squares Value Iteration (RLSVI) (Osband et al., 2016b) provided an early analytical connection, showing that bootstrap-style perturbations can approximate Gaussian posteriors and achieve polynomial sample efficiency. Osband et al. (2016a) extended this concept to deep neural networks. The method attaches a separate output head (a linear layer) for each of the $K$ Q-value estimates and uses masks so that gradients for each head flow only through a specific subset of data mini-batches. This results in an ensemble of DQNs whose stationary weight distribution can approximate a Bayesian posterior, particularly in the limit of small learning rates (Mandt et al., 2017). In general, however, this method does not necessarily approximate a Bayesian posterior; notably, this does not make it less useful for efficient exploration. Each ensemble member is trained on its own target network, allowing for the correct propagation of time-discounted uncertainties (Osband et al., 2016a).

**Injecting priors.** Despite its empirical success, vanilla BDQN has a limitation: its belief about the environment is formed from the *same* training data for all the models. Consequently, in regions of the state-action space that the agent has not visited, its uncertainty can diminish too quickly due to the alignment of training gradients for different ensemble members. To address this, Osband et al. (2018) proposed adding a frozen, untrainable *prior network* to each member of the ensemble. The resulting *Randomized-Prior BDQN* (RP-BDQN) maintains computational tractability while restoring a more robust (non-degenerate) posterior, and it maintains theoretical guarantees under linear function approximation. Meng et al. (2023) attempted to replace the priors with white noise; however, we were unable to reproduce their results. Therefore, our work focuses on the canonical and widely adopted RP-BDQN formulation. In RP-BDQN, each of the $K$ members (with parameters $\theta_k$ and frozen prior $p_k$) has a total predicted value $\hat{Q}_k(s,a) = Q_k(s,a;\theta_k) + p_k(s,a)$. The ensemble is trained to minimize the following loss:

$$L(\theta) = \frac{1}{K} \sum_{k=1}^{K} \mathbb{E}_{(s,a,r,s')\sim\mathcal{D}} \left[ (y_k - \hat{Q}_k(s,a))^2 \right]$$
$$\text{where} \quad y_k = r + \gamma \max_{a'} \hat{Q}_k(s',a';\theta_k^-).$$

**Towards scaling laws for exploration.** While BDQN and RP-BDQN are widely used and have demonstrated strong empirical performance, a systematic understanding of *how* their effectiveness scales with factors like task difficulty and ensemble size is still lacking. Drawing inspiration from the power-law tradition observed in fields like language modeling (Kaplan et al., 2020) and supervised deep learning (Hestness et al., 2017), our goal is to identify simple empirical regularities that can help practitioners make informed decisions about resource allocation when using these methods in deep RL.

## 3 Experimental Setup

To derive empirical scaling laws for exploration, we must first construct a methodology that allows us to (1) precisely control the difficulty of the exploration problem and (2) robustly measure an agent's ability to solve it. This section outlines our approach, detailing the environment used to parameterize exploration hardness, the metric used to quantify success, the specific agents subjected to this test, and the quantitative models that emerge from the results.

### 3.1 Measuring Exploration in a Parameterized Environment

To isolate and measure exploration capability, we use the **DeepSea**($n$) environment (Osband et al., 2018), an $n \times n$ grid world where a single rewarded state is reachable only via a long, penalized path (Fig. 2). The grid size $n$ serves as a direct, tunable **hardness parameter**. Our performance metric is the **Probability of Discovery (PoD)**: the fraction of training runs (over different random seeds) where the agent finds the goal at least once within a fixed step budget. This metric directly quantifies an algorithm's ability to solve the exploration problem, independent of its subsequent policy optimization speed.

We deliberately chose DeepSea as our primary benchmark because it uniquely provides a clean and interpretable "hardness" parameter ($n$), allowing us to systematically vary and quantify exploration difficulty. In contrast, most popular RL environments do not offer a transferable or well-defined metric for exploration complexity, making it infeasible to disentangle the effects of algorithmic changes from environmental confounds. DeepSea's controlled setting thus enables precise investigation of scaling laws and exploration dynamics, which would be otherwise obscured in more complex domains.

### 3.2 Agents and Implementation Details

We use our testbed to compare the scaling properties of two key posterior sampling approximations: BDQN and RP-BDQN.

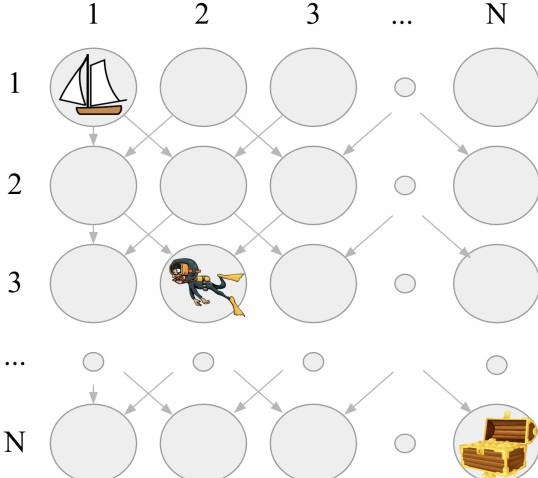

Figure 2: **DeepSea**($n$) **environment.** The agent starts at the top-left corner; the only rewarding state is the bottom-right corner, and any right move incurs a small penalty, while any left move yields no reward. Larger grids require proportionally longer penalized action sequences, making exploration harder. Diagram adapted from Osband et al. (2020).

- **Bootstrapped DQN (BDQN):** An ensemble of $K$ DQN networks. Our implementation, which does not use a shared network 'core,' is more aligned with the RP-BDQN architecture and can be viewed as a no-prior variant of Osband et al. (2018), with similar performance to that of Osband et al. (2016a).

- **Randomized-Prior BDQN (RP-BDQN):** An exact reproduction of Osband et al. (2018), which extends BDQN by adding the output of a fixed, randomly initialized prior network to each of the $K$ separate ensemble members.

**Environment and Compute.** All agents are implemented in JAX (Bradbury et al., 2018) and Equinox (Kidger & Garcia, 2021). Our DeepSea implementation is based on Gymnax (Lange, 2022) with minor corrections, as described in Appendix, enabling fully GPU-accelerated training. To ensure computational equivalence, we scale both the learning rate and batch size linearly with ensemble size $K$. This is required to maintain the same first and second moments of the per-member updates for different ensemble sizes. For completeness, we include the derivation of this fact in the Appendix. The replay buffer size is held constant, and all ensemble members share transitions. For each algorithm and each pair $(n, K)$, we perform 32 training runs using different seeds, with a limit of 50,000 episodes per run.

**Network Architecture and Hyperparameters.** To ensure a fair comparison, all agents use the same underlying Q-network and prior network architecture: a Multi-Layer Perceptron (MLP) with two hidden layers of 32 units and GLU activation function (Dauphin et al., 2017). Most of the hyperparameters are held constant across all experiments and are listed in Table 1. We train our model using the AdamW (Loshchilov & Hutter, 2019) optimizer, without learning rate decay.

### 3.3 Experimental Protocol

Our investigation proceeds from a large-scale grid search in DeepSea across the ensemble size $K \in \{1, 2, 3, 4, 6, 10, 16, 20, 24, 32, 40\}$. For each algorithm and value of $K$, we start from $n = 3$ and increase it until we observe zero successful runs (out of 32 random seeds) for three consecutive values of $n$.

Table 1: Default hyperparameters used for all agents, with $K$ corresponding to ensemble size.

| Hyperparameter name | Value |
|---|---|
| Learning rate | 3e-4$\times K$ |
| Weight decay | 1e-6 |
| Replay buffer size | 10,000 |
| Discount factor ($\gamma$) | 0.99 |
| Batch size | $128 \times K$ |
| Target network update frequency | 500 steps |
| Prior scale (for RP-BDQN) | 3.0 |

## 4 An Effective Scaling Model

What is the global performance landscape for each method? We visualize the Probability of Discovery as a function of both task hardness ($n$) and ensemble size ($K$). Figure 3 presents these results as a heatmap and an efficiency frontier.

For both BDQN and RP-BDQN, increasing the ensemble size yields diminishing returns, offering only a modest extension of the solvability frontier past $K = 20$. However, even a small RP-BDQN ensemble can solve problems that are entirely intractable for a BDQN ensemble 5 times larger. We note that using Randomized Priors increases the maximum solvable hardness of the problem by a factor of 1.5 using the hyperparameter configuration specified in Table 1.

### 4.1 A First-Order Model of Discovery

The observed empirical trends can be described by the following simple, effective model:

$$P(\text{discovery}) \approx 1 - (1 - \psi^n)^K \tag{1}$$

This mathematical form has a direct interpretation as a 'best-of-K' independent trials model. It posits that an ensemble succeeds if at least one of its members solves the task, assuming each member has an independent probability of success $\psi^n$. The goal of using such a deliberately simple model is not to claim it represents the ground truth, but to test how much of the complex ensemble behavior can be explained by the most basic possible assumption: that the members do not interact.

We fit this model to our experimental data for BDQN and RP-BDQN by maximizing the log-likelihood. The resulting parameters and goodness-of-fit metrics are shown in Table 2, with the fitted equiprobability curves plotted in the bottom row of Figure 3.

Table 2: Fitted parameters and goodness-of-fit for the unified scaling law. The parameter $\psi$ represents the base effectiveness of a single ensemble member. The goodness-of-fit ($R^2$) measures the proportion of variance in the mean performance explained by the model. The dispersion is estimated as the Pearson chi-squared statistic ($\chi^2$) divided by the residual degrees of freedom ($df$). High dispersion values ($> 1$) indicate that significant unmodeled dynamics (e.g., correlations) remain, confirming the model is a first-order approximation. Values after $\pm$ represent 95% confidence intervals computed via bootstrap.

| Parameter | BDQN | RP-BDQN |
|---|---|---|
| Parameter ($\psi$) | $0.80 \pm 0.02$ | $0.87 \pm 0.01$ |
| Goodness-of-fit ($R^2$) | 0.84 | 0.69 |
| Dispersion | 4.1 | 8.1 |
| MSE | 0.024 | 0.049 |

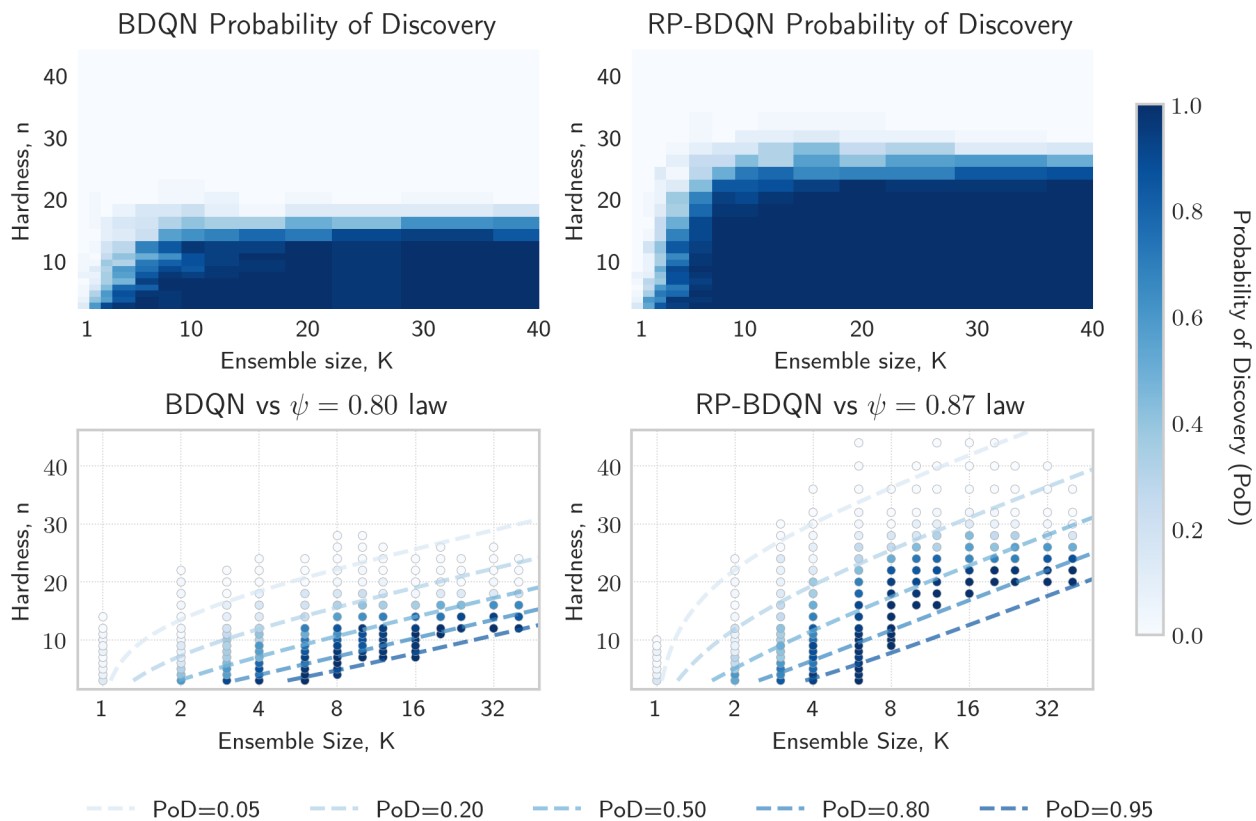

Figure 3: **Scaling properties of BDQN and RP-BDQN.** The probability of discovery (PoD) is the effective probability that at least one agent in an ensemble of size $K$ solves a problem of hardness $n$. **Top row**: Empirical PoD for BDQN (left) and RP-BDQN (right). **Bottom row**: Raw experimental data, where each point is colored according to its PoD. The horizontal axis is logarithmic in ensemble size. The dashed lines are contours of constant PoD from the fitted model, $p = 1 - (1 - \psi^n)^K$, with $\psi = 0.80$ for BDQN and $\psi = 0.87$ for RP-BDQN. The model accurately captures the general scaling behavior, although a mismatch for large $K$ can be observed: the $n \geq 32, K = 32$ interval should have a probability of 0.4 according to the RP-BDQN model but has zero empirical discoveries across all 128 runs. Note that the compute spent on an experiment is directly proportional to K, see more in Appendix B.

## 4.2 Model Interpretation and Deviations as Diagnostics

The model's high $R^2$ value of approximately 0.8 is noteworthy. It indicates that a model assuming no cooperation or shared information between ensemble members can still explain the majority of the variance in performance. This suggests that, to a first order, the primary benefit of ensembling in this context comes from having multiple independent chances at discovery, rather than from complex emergent cooperation.

However, the model is an approximation, and its failures are diagnostically useful. The residual plot in Figure 4 reveals a distinct U-shaped structure. These systematic deviations from the independent agent model allow us to identify and mechanistically explain higher-order dynamics:

- **Small ensembles ($K < 4$):** The model overestimates performance (residuals $> 0$). Since we fit a single $\psi$ across all ensemble sizes, this likely reflects that small ensembles have limited initial diversity, making the globally-fitted $\psi$ too optimistic for this regime. Notably, the residual at $K = 1$ (greedy DQN) is not strongly negative, which would have implied that running $N$ independent

greedy DQNs outperforms a $N$-member bootstrap ensemble, suggesting the ensemble structure does provide benefit beyond independent trials.

- **Matching regime** ($4 \leq K \leq 20$)**:** The predicted probability of discovery matches the empirical one well. In this range, the independent-trials assumption holds approximately, and the shared replay buffer may provide mild implicit cooperation: when one member discovers the goal, others can learn from that transition without independent exploration.

- **Correlated Failures and Saturation** ($K > 20$)**:** The model again overestimates performance (residuals $> 0$), implying diminishing returns from additional members. We attribute this to **diversity saturation**: even with randomized priors, the number of distinct basins of attraction in the loss landscape is finite. As $K$ grows, multiple members inevitably converge to similar local optima, leading to correlated failures.

In summary, this simple model serves two purposes: it provides a good first-order approximation of performance, and its systematic failures provide a quantitative window into the more complex, underlying ensemble dynamics.

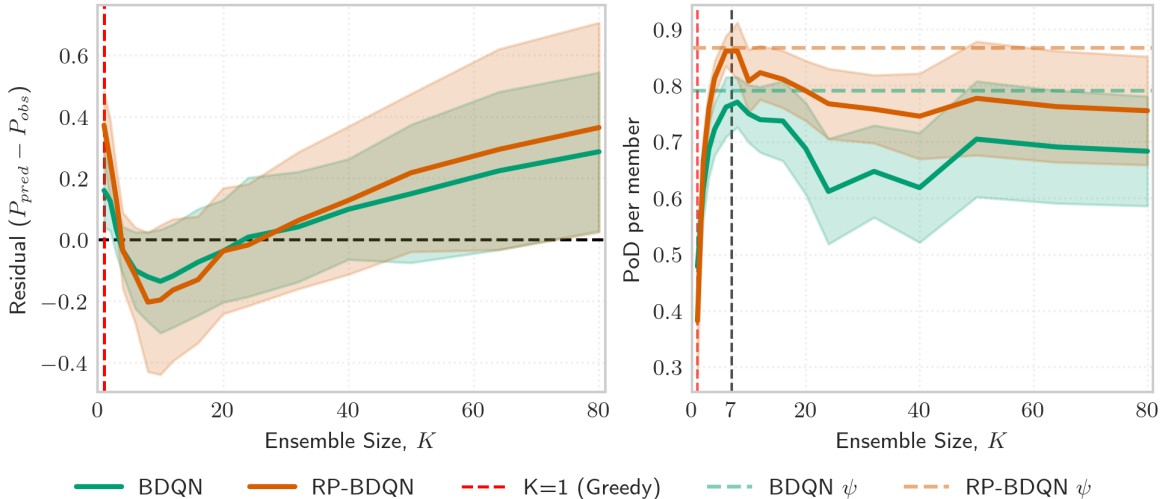

Figure 4: **Residuals of the Scaling Model and Efficiency. Left**: The prediction error (Model PoD - Empirical PoD) versus ensemble size $K$. The U-shape reveals systematic model failures: overestimating performance for both small and large $K$, but predicting the PoD well for ensemble size in a reasonable range $4 < K < 20$. **Right:** The PoD per unit of compute versus ensemble size. The curve peaks at small ensemble sizes, $K = 7$, showing that using larger ensemble sizes has no utility.

### 4.3 Computational Trade-offs

Since our implementation preserves the first and second moments of gradient updates per member (see Appendix B), the computational cost (FLOPs) scales linearly with $K$. We define *computational efficiency* as the probability of discovery per unit of compute (PoD per member $\propto$ PoD/FLOPs).

As illustrated on the right subplot of Figure 4, this efficiency metric is a curve with a distinct peak - surprisingly for both BDQN and RP-BDQN - around $K = 7$, afterwards declining slightly.

This saturation phenomenon has a simple implication for compute allocation. As shown in Section 4.2, large ensembles ($K > 20$) suffer from correlated failures (diversity saturation), performing worse than the independent scaling law predicts. By splitting a large compute budget into independent restarts (i.e. running two separate ensembles of size $K = 10$ rather than one of $K = 20$), practitioners can structurally enforce independence. This effectively resets the diversity landscape, making this strategy compute-wise superior.

### 4.4 The Underlying Mechanism: Ensemble Diversity

The performance heatmaps and scaling models demonstrate that RP-BDQN is substantially more effective than BDQN, but they do not explain why. We hypothesize that the performance gap stems directly from RP-BDQN's ability to maintain a diverse set of exploratory policies, whereas vanilla BDQN is prone to premature posterior collapse, where all members converge to a single, often suboptimal, policy.

To test this hypothesis, we need a quantitative measure of ensemble diversity. While diversity in the high-dimensional parameter space is difficult to track, we can readily measure it in the function space of Q-values. We propose the following metric for **Q-Diversity**:

1. At the beginning of each training run, we sample a fixed set of 64 probe states, $\mathcal{S}_{probe}$, from the environment.

2. Periodically during training, for each state $s \in \mathcal{S}_{probe}$, we compute the Q-values, $Q_k(s, a)$, for all actions $a$ and for each of the $K$ ensemble members.

3. The Q-Diversity at a given time is the standard deviation of these Q-values across the ensemble members, averaged over all 64 probe states.

A high Q-Diversity value indicates that the ensemble members disagree on the values of actions, suggesting they are pursuing different exploratory strategies. A low value signifies consensus and a collapse in exploration.

We analyze this metric in the most informative regime: a region of task difficulty where success is uncertain ($8 \leq n \leq 12$ and $3 \leq K \leq 6$). We partition the runs from this region into two groups: 'convergent' (those that successfully find the goal) and 'non-convergent' (those that do not). Figure 5 plots the evolution of the average Q-Diversity for these groups.

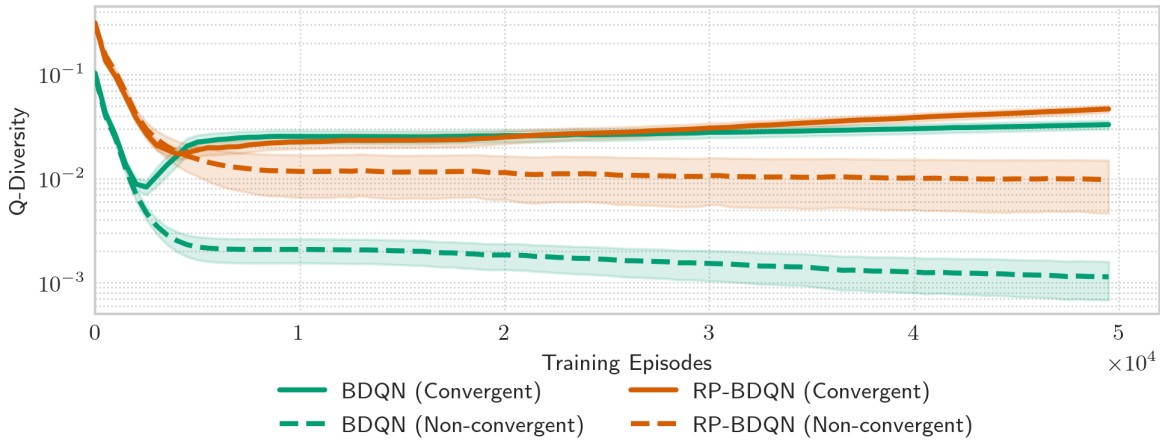

Figure 5: **Ensemble Diversity Collapse During Training.** Evolution of Q-Diversity in a challenging regime ($PoD \approx 0.5$). For BDQN, diversity collapses prematurely in non-convergent runs. RP-BDQN maintains significantly higher diversity, explaining its superior exploration. Shaded regions show the standard deviation across runs.

The results provide clear, empirical evidence for our hypothesis. For RP-BDQN, Q-Diversity remains high throughout the training process; the randomized priors effectively prevent the ensemble from collapsing. Runs that eventually converge maintain high diversity for a significant portion of the training, only decreasing after the optimal policy is likely to be found. In contrast, the BDQN runs show a premature drop in diversity, and the non-convergent runs exhibit tenfold lower variability than RP-BDQN counterparts. This indicates that the ensemble members quickly agree on a suboptimal greedy policy, effectively ending meaningful exploration and trapping the agent in a suboptimal solution.

This analysis confirms that the superior performance of RP-BDQN is not merely an incidental benefit but likely a consequence of its mechanism: the enforced preservation of ensemble diversity.

### 4.5 Robustness Across Hyperparameters

An important question is whether our proposed scaling model and the observed performance gap between BDQN and RP-BDQN are merely artifacts of our chosen hyperparameters or if they represent a more general principle. To validate the robustness of our findings, we conducted a series of hyperparameter sweeps around our default configuration. We systematically varied the learning rate $\eta \in \{8 \cdot 10^{-5}, 5 \cdot 10^{-4}, 10^{-3}\}$, replay buffer size $|\mathcal{D}| \in \{5000, 20000, 40000\}$, and, for RP-BDQN, the prior scale $\beta \in \{1.0, 5.0, 10.0\}$.

For each new hyperparameter configuration, we re-ran the entire evaluation across the lower-resolution range of task hardness ($n$) and ensemble size ($K$) and re-fit our scaling model (Eq. 1) to determine the effective *best-of-one* success parameter, $\psi$. This allows us to measure how the fundamental effectiveness of each algorithm changes rather than just observing performance on a single task. The results, summarized in Figure 6, demonstrate the remarkable robustness of our central finding.

While the choice of hyperparameters modulates the absolute value of $\psi$, the $\psi$ for RP-BDQN remains consistently higher than that of BDQN. We observe several secondary trends:

- **Learning Rate and Replay Buffer:** For both algorithms, performance is sensitive to these standard hyperparameters. However, the scaling law parameter is *not* sensitive, which shows that the scaling law is robust with respect to these hyperparameters.

- **Prior Scale:** The prior scale controls the magnitude of the randomized prior's contribution and has a notable impact on RP-BDQN's performance. Our default value of 3.0 is already strong, but the sweep suggests that, in general, increasing this prior scale improves performance. The prior work of Osband et al. (2018) suggested that a prior scale of 5 could be optimal. However, we find no saturation of performance past this value. This could be attributed to the slight differences in our training pipeline.

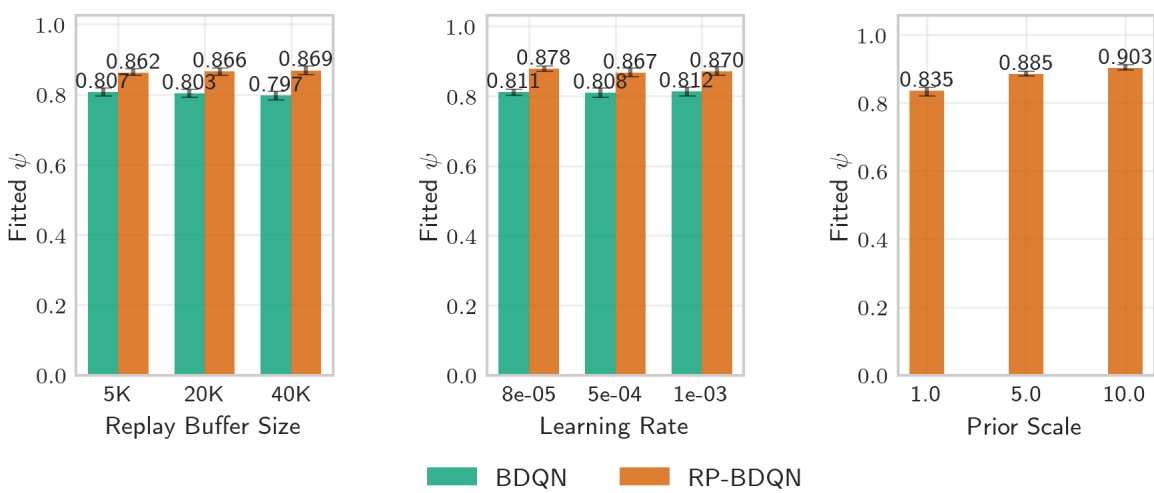

Figure 6: **Robustness of the Scaling Model Parameter $\psi$ Across Hyperparameters.** The fitted *best-of-one* success parameter, $\psi$, is plotted for sweeps over (a) replay buffer size, (b) learning rate, and (c) prior scale. The error bars represent the 95% confidence interval from the model fit.

## 5 Discussion

Our study intentionally utilizes DeepSea to provide a precise, parameterized lens on exploration scaling. By doing so, we isolate the algorithmic contribution from environmental confounds. However, this raises the question of transferability to more complex domains like Atari or continuous control.

### 5.1 Generalizing Hardness and the Scaling Law

In unstructured environments, hardness $n$ is not an explicit parameter but an emergent property of the optimization landscape, transition stochasticity, and horizon length. While one might consider proxy metrics, such as the expected number of steps to a first reward under a random policy, we believe that such proxy metrics often conflate pure exploration difficulty with policy optimization challenges.

Therefore, we do not propose a single universal proxy for the hardness. While the specific value of $\psi$ will vary by domain, we posit that the structural form of our scaling law $(1 - (1 - p)^K)$ will remain a robust baseline. This would allow practitioners to use the model diagnostically: measuring performance against this curve reveals whether a method is benefiting from cooperation (beating the curve) or suffering from diversity collapse (falling below it).

### 5.2 A Prescriptive Guide for Practitioners

Our diagnostic findings transform the scaling model from a descriptive tool into a prescriptive one. Deviations from the "best-of-K" curve provide actionable guidance:

1. **Diagnosing Diversity Saturation:** If performance falls *below* the baseline (the right side of the U-curve in Fig. 4), the ensemble members are failing in correlated ways. This indicates that the optimization landscape's capacity for distinct local minima has been saturated. In this regime, computational resources increasing the ensemble size is less beneficial, and one might want to focus on the alternative ways to preserve diversity.

2. **Leveraging Information Sharing:** Performance exceeding the baseline suggests implicit cooperation via the shared buffer; practitioners can amplify this by prioritizing replay of high-error transitions found by peers.

## 6 Conclusions and Future Work

We presented a simple, effective model that describes reward discovery as a function of task hardness and ensemble size, governed by an effectiveness parameter, $\psi$, in deterministic, sparse-reward settings. We showed that RP-BDQN is substantially more effective than BDQN due to its ability to maintain ensemble diversity, and we offered practical guidance, noting that returns diminish for ensemble sizes $K > 10$.

Our findings open several avenues for future research:

- **Optimizing for $\psi$:** Moving from a descriptive to a prescriptive use of our scaling model, for instance, by designing algorithms or regularization schemes that explicitly aim to maximize $\psi$.

- **Refining the Effective Model:** Developing a more nuanced model that incorporates terms to account for the cooperative effects at small $K$ and the correlated failures at large $K$, thereby better capturing the diagnosed second-order ensemble dynamics.

- **Stochastic and Continuous Domains:** Future work should validate whether the structural form of the scaling law holds in environments with stochastic transitions or continuous action spaces.

By establishing a quantitative framework for scaling, we hope to advance the field toward more predictable and practical exploration algorithms.

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

## Appendix

## A    Environment implementation.

Our study required a high-performance implementation of the DeepSea environment to facilitate large-scale experimentation. We chose to build upon the JAX-based `gymnax` library (Lange, 2022) to enable fully GPU-accelerated training and environment stepping, thereby minimizing CPU-GPU data transfer overhead.

During initial validation, we observed anomalously high performance, with agents solving tasks of hardness $n = 100$ with small ensemble sizes. An investigation revealed that the existing `gymnax` implementation had the randomized action map disabled by default - a "debug" configuration from the original environment specification (Osband et al., 2020). We corrected this discrepancy to enforce the proper exploration challenge. To guarantee correctness, we verified our fixed implementation against the original by running approximately $20,000$ in automated tests confirming trajectory-wise reward equivalence between the two versions.

**Performance Optimization.**    To maximize memory bandwidth and training throughput, we implemented a transparent observation compression mechanism within our replay buffer. While the agent interacts with standard $(n, n)$-shaped observation tensors, the buffer internally represents each observation as a single integer. This optimization resulted in a threefold improvement in overall training performance.

## B    Consistent moments of the per-member gradients.

### B.1    Implementation: Averaged Loss over Split Batches

For each optimization step:

1. A total data batch of size $B_{\text{total}} = K \cdot B_1$ is sampled, where $K$ is the ensemble size and $B_1$ is the baseline batch size.

2. This batch is split into $K$ unique, non-overlapping mini-batches $(D_1, D_2, \ldots, D_K)$, each of size $B_1$.

3. Each ensemble member $k$, with its independent parameters $\theta_k$, calculates its loss $\ell_k(\theta_k; D_k)$ on its corresponding mini-batch $D_k$.

4. The final loss function to be differentiated is the average of these individual losses: $L_{\text{avg}} = \frac{1}{K} \sum_{j=1}^{K} \ell_j(\theta_j; D_j)$.

5. The learning rate is scaled linearly: $\eta_K = K \cdot \eta_1$.

## B.2 Derivation of the Per-Member Gradient

The key insight comes from calculating the gradient of the total average loss $L_{\text{avg}}$ with respect to the parameters $\theta_k$ of a single, specific member $k$. Because the members do not share parameters, the loss of member $j$ ($\ell_j$) does not depend on the parameters of member $k$ for any $j \neq k$. Therefore, $\nabla_{\theta_k} \ell_j(\theta_j; D_j) = 0$ for all $j \neq k$.

The gradient for member $k$ is:

$$
\begin{aligned}
\nabla_{\theta_k} L_{\text{avg}} &= \nabla_{\theta_k} \left( \frac{1}{K} \sum_{j=1}^{K} \ell_j(\theta_j; D_j) \right) \\
&= \frac{1}{K} \sum_{j=1}^{K} \nabla_{\theta_k} \ell_j(\theta_j; D_j) \\
&= \frac{1}{K} \left( \nabla_{\theta_k} \ell_1 + \cdots + \nabla_{\theta_k} \ell_k + \cdots + \nabla_{\theta_k} \ell_K \right) \\
&= \frac{1}{K} \nabla_{\theta_k} \ell_k(\theta_k; D_k)
\end{aligned}
$$

Let $g_k = \nabla_{\theta_k} \ell_k(\theta_k; D_k)$ be the gradient for member $k$ on its own mini-batch. The gradient used by the optimizer for member $k$'s parameters is simply $\frac{1}{K} g_k$.

## B.3 Analysis of the Per-Member Parameter Update

The parameter update $\Delta\theta_k$ for member $k$ is calculated using the scaled learning rate $\eta_K$ and its specific gradient component $\frac{1}{K} g_k$:

$$
\Delta\theta_k = \eta_K \cdot \left( \frac{1}{K} g_k \right)
$$

Substituting the scaled learning rate $\eta_K = K\eta_1$:

$$
\Delta\theta_k = (K\eta_1) \cdot \left( \frac{1}{K} g_k \right) = \eta_1 g_k
$$

This is a powerful result: the scaling of the learning rate by $K$ and the down-scaling from the averaged loss by $1/K$ perfectly cancel. The resulting update rule for an ensemble member, $\Delta\theta_k = \eta_1 g_k$, is identical in form to the update rule for a baseline single agent, where $g_k$ is computed on a batch of size $B_1$.

## B.4 Moment Equivalence

We can now confirm that the moments of the per-step update are identical to the baseline case.

- **First Moment (Mean):** The expectation of the update for an ensemble member is:

$$
\mathbb{E}[\Delta\theta_k] = \mathbb{E}[\eta_1 g_k] = \eta_1 \mu_k
$$

This is identical to the baseline agent's mean update.

- **Second Moment (Covariance):** The covariance of the update for an ensemble member is:

$$\text{Cov}(\Delta\theta_k) = \text{Cov}(\eta_1 g_k) = \eta_1^2 \text{Cov}(g_k) = \frac{\eta_1^2}{B_1}\Sigma_k$$

This is identical to the baseline agent's update covariance.

## B.5 Conclusion

The "split batch" implementation with a scaled learning rate ensures that the statistical properties (mean and covariance) of the per-step parameter update for each ensemble member are identical to those of a single agent trained with baseline hyperparameters $(B_1, \eta_1)$. The total batch size is scaled to $K \cdot B_1$ to make efficient use of parallel hardware, allowing $K$ models to be trained for the wall-clock time of one, thus achieving a $K$-fold speedup in terms of data processed over time. This methodology provides a foundation for achieving computationally equivalent training across different ensemble sizes.

