# OpenReview forum: "A Simple Scaling Model for Bootstrapped DQN"
_TMLR — Rejected by TMLR_

### Review · Reviewer_2xR9 · 2025-12-05

**Summary Of Contributions:**

This paper presents a large-scale empirical analysis of Bootstrapped DQN (BDQN) and Randomized-Prior BDQN (RP-BDQN) within the DeepSea environment. The primary contribution is a simple scaling model that accurately predicts the Probability of Discovery (PoD) as a function of task hardness ($n$) and ensemble size ($K$): $P(\text{discovery}) \approx 1 - (1 - \psi^n)^K$.

Strengths:
1. Quantitative Comparison: The model enables a direct, quantitative comparison of the two methods through the effectiveness factor ($\psi$), showing RP-BDQN ($\psi \approx 0.87$) is substantially more effective than BDQN ($\psi \approx 0.80$). This advantage is attributed to RP-BDQN's ability to sustain ensemble diversity and mitigate posterior collapse.
2. Diagnostic Framework: The study uses systematic deviations from the simple "best-of-K" model to diagnose complex, second-order effects: cooperation in small ensembles and diversity saturation/correlated failures in large ensembles.

Weakness:
1. Limited Scope: The entire analysis is confined to the DeepSea environment and only two algorithms, raising questions about the generalizability of the specific scaling parameters to more complex, real-world RL domains (e.g., Atari).

**Additional Comments:**

Nope

**Audience:**

Yes

**Audience Explanation:**

The paper might interest researchers in Deep Reinforcement Learning (RL), who focus on epistemic exploration and ensemble methods.

**Broader Impact Concerns:**

Nope

**Claims And Evidence:**

Yes

**Claims Explanation:**

Most claims are supported by systematic empirical studies

**Requested Changes:**

1. **Limited Scope and Transferable Guidance:** The analysis focuses exclusively on two algorithms (BDQN and RP-BDQN) within a single, abstract environment (DeepSea). While the authors justify this setup as a "laboratory" for isolating exploration effects, the resulting scaling model lacks the predictive, transferable power of similar scaling laws established in other domains (e.g., LLM scaling laws [a], which provide quantitative guidance on optimal ratios between parameters, data, and computation).
    * The paper currently provides specific numbers ($\psi \approx 0.87$, optimal $K \approx 10$) that are likely confined to DeepSea. To mitigate the perception of limited applicability, the authors should explicitly discuss which elements of the scaling law framework are expected to generalize (e.g., the structural form $1 - (1 - \psi^n)^K$) and which are environment-specific (the parameter $\psi$).
    * The discussion should argue more forcefully for how this methodology, rather than the specific numerical values, can be applied to inform algorithm design in more complex, real-world RL domains.
2. **Depth of Interpretation vs. Naming of Observation:** The interpretation of systematic model deviations—such as "cooperative effects" at small ensemble sizes ($K$) and "diversity saturation/correlated failures" at large $K$—is currently a diagnostic observation (i.e., naming the pattern of the model's failure). It is not yet a causal explanation.
    * To benefit future work, the authors should move beyond merely naming these effects. They should propose and discuss the underlying architectural or algorithmic mechanisms driving these phenomena. For instance, what is the mechanism by which the shared replay buffer specifically creates a "cooperative effect,” and how could an algorithm be modified to enhance this beneficial effect?
    * The authors should provide a prescriptive example of how these diagnostic findings (the U-shaped residual) could benefit a future researcher. Without such prescriptive guidance, the interpretation remains a relabeling of empirical results.

[a] Scaling Laws for Neural Language Models, Arxiv 2020

---

> ### Author Response · Authors · 2026-01-07
>
> We thank the reviewer for their constructive feedback and for identifying the value of our diagnostic framework. We agree that moving from a descriptive model to prescriptive guidance is crucial for the broader applicability of this work.
>
> Regarding the limited scope and transferability, we have expanded Section 5.1 to distinguish between environment-specific parameters and the generalizable framework. We posit that the structural form of the scaling law acts as a robust theoretical baseline for independent search across domains. This allows the methodology to serve as a diagnostic tool: practitioners can plot performance against this curve to determine if a method is effectively utilizing cooperation or suffering from mode collapse, regardless of the specific environment.
>
> Regarding the depth of interpretation, we refined Section 4.2 and added a new Section 5.2 to address this. We explicitly link the "cooperative" mechanism to the shared replay buffer, which acts as a communication channel for high-reward transitions found by single members. We translated these diagnostics into actionable advice in the new section; for instance, we advise that if performance falls below the scaling baseline (saturation), practitioners should structurally enforce independence via restarts rather than simply increasing the ensemble size.

---

> > ### Comment · Reviewer_2xR9 · 2026-01-26
> >
> > Thank you to the authors for the response, which addresses some of my concerns. However, I still have a follow-up question.
> >
> > In the rebuttal, the authors state that when K < 4 the residual is negative, while when K > 20 the residual becomes positive. However, based on Figure 4, the residuals in both regions appear to be positive, which seems inconsistent with this description. In addition, the y-axis is labeled as $P_{\text{pred}} - P_{\text{obs}}$, whereas the caption states “Empirical PoD - Model PoD,” which suggests the opposite sign. Could the authors clarify this discrepancy and confirm the correct definition of the residual used in Figure 4?

---

> ### Author Response · Authors · 2026-02-04
>
> Thank you for pointing this out.
>
> You are correct that there was an inconsistency between the textual description and Figure 4. In the original version, the plot (including the y-axis definition of the residual as $P_{\text{pred}} - P_{\text{obs}}$) was correct, while the accompanying text describing the sign of the residual for different values of $K$ was incorrect. We have rewritten this section and updated the caption to ensure that the definition of the residual and its sign are stated clearly and consistently throughout.
>
> Thank you for bringing this discrepancy to our attention.

---

### Review · Reviewer_wnpu · 2025-12-09

**Summary Of Contributions:**

### **Summary Of Contributions**

The authors present a large-scale empirical study focused on characterizing the exploration performance of Bootstrapped DQN (BDQN) and Randomized-Prior BDQN (RP-BDQN). Using the `DeepSea` environment to precisely parameterize exploration difficulty (hardness $n$), the paper proposes a simple "best-of-K" scaling model: $P(\text{discovery}) \approx 1 - (1 - \psi^n)^K$.

**Key Contributions:**
1.  **Scaling Law:** The derivation and empirical fitting of a scaling model that unifies task hardness and ensemble size. The model identifies an effectiveness parameter $\psi$, showing RP-BDQN ($\psi \approx 0.87$) consistently outperforms BDQN ($\psi \approx 0.80$).
2.  **Mechanism Analysis:** The paper introduces "Q-Diversity" as a metric, demonstrating that RP-BDQN's success stems from sustained ensemble diversity, whereas BDQN suffers from premature posterior collapse.
3.  **Diagnostic Framework:** The authors analyze systematic deviations (residuals) from their model to diagnose second-order dynamics, specifically identifying cooperative effects at small ensemble sizes ($K < 4$) and diversity saturation/correlated failures at large ensemble sizes ($K > 20$).
4. **Technical Implementation:** A rigorous validation of ensemble training equivalence (scaling learning rates linearly with $K$ to maintain moment consistency) and a correction to the standard JAX `DeepSea` implementation.

**Strengths:**
* **Controlled Experimentation:** The use of `DeepSea` allows for a clean isolation of exploration difficulty, which is often conflated with other factors in environments like Atari.
* **Rigorous Methodology:** The derivation of the "split batch" gradient moments in Appendix C is excellent and ensures fair comparison across ensemble sizes.
* **Honest Analysis:** The paper explicitly analyzes where their simple model fails (the U-shaped residuals), using these failures to provide deeper insight rather than glossing over them.

**Weakness:**
* **Environment Specificity:** The empirical validation is limited to `DeepSea`. While justified for parameterization purposes, the transferability of the specific $\psi$ values or the precise curve shape to stochastic or continuous domains is unknown.

**Additional Comments:**

I would like to commend the authors for Appendix C ("Consistent moments of the per-member gradients"). In ensemble RL literature, it is very common to see implementations that simply sum losses without scaling learning rates, leading to unfair comparisons where larger ensembles effectively have higher learning rates. Deriving and enforcing the condition $\Delta \theta_k = \eta_1 g_k$ is a strong indicator of technical quality. Similarly, catching the `gymnax` bug likely saved the community from future non-reproducible baselines. This technical rigor strongly supports acceptance.

**Audience:**

Yes

**Audience Explanation:**

The paper addresses fundamental questions in Deep Reinforcement Learning regarding exploration and ensemble methods.
1.  **Practitioners:** The finding that performance benefits diminish significantly for $K > 10$ is practically useful for resource allocation in expensive training setups.
2.  **Theorists:** The scaling law provides a "clean" empirical baseline for deep exploration, akin to scaling laws in LLMs, which invites further theoretical work on why independent agent assumptions hold so well despite shared buffers.
3.  **Methodology:** The rigor regarding JAX implementation and gradient scaling is valuable for the growing community of researchers using JAX for RL.

**Broader Impact Concerns:**

No concerns. The paper is a fundamental study of algorithmic efficiency and exploration in reinforcement learning. It does not present direct ethical risks.

**Claims And Evidence:**

Yes

**Claims Explanation:**

The submission provides robust evidence for its claims:

1. **Scaling Model Fit:** The authors performed over 40,000 experimental runs. The fit of the proposed model is quantitatively supported by high $R^2$ values ($0.84$ for BDQN and $0.69$ for RP-BDQN) and visual alignment with the efficiency frontiers in Figure 3.
2. **Performance Gap:** The superiority of RP-BDQN is clearly established through the heatmaps in Figure 3 and the robustness sweeps in Figure 6, which show RP-BDQN maintaining a higher $\psi$ across varying learning rates and buffer sizes.
3.  **Diversity Mechanism:** Figure 5 provides convincing evidence for the mechanism claim. It clearly shows that "non-convergent" BDQN runs exhibit a collapse in Q-diversity, while RP-BDQN maintains high diversity, directly correlating diversity with the probability of discovery.
4.  **Implementation Correctness:** The authors address potential confounds by mathematically proving the equivalence of their update rules for varying $K$ in Appendix C, and they identified/fixed a critical bug in the `gymnax` library that would have otherwise invalidated the results.

**Requested Changes:**

**1. Address Environment Specificity and Transferability (Strengthen)**
The empirical validation is strictly limited to `DeepSea`, which is a deterministic, discrete grid world. While the paper justifies this choice to isolate exploration hardness, the transferability of the specific $\psi$ values or the precise curve shape to stochastic or continuous domains remains unknown. Please expand the Discussion section to explicitly analyze the limitations regarding transferability. Specifically, discuss how the scaling law $1 - (1 - \psi^n)^K$ might behave in environments with stochastic transitions or continuous action spaces. Is "Hardness $n$" solely a function of horizon, or would it effectively increase with stochasticity? Providing a structured hypothesis here would add significant value for researchers looking to apply this framework to more complex settings like Atari.

**2. Elaboration on "Cooperative Effects" (Strengthen):**
The paper notes that for $K < 4$, the empirical performance is *better* than the independent model predicts, attributing this to "cooperation" via the shared replay buffer. Please expand slightly on this hypothesis. Is there a way to verify this? For example, does this positive residual disappear if the replay buffers were separated (conceptually)? A brief discussion (even speculative) in Section 4.2 would be valuable to clarify *how* the shared buffer induces cooperation (e.g., one agent finding a reward puts it in the buffer, allowing others to train on it immediately).

**3. Discussion on Generalization of "Hardness" (Strengthen):**
The paper correctly identifies the reliance on `DeepSea` as a limitation. In the Discussion/Conclusion, please expand on how a practitioner might estimate "Hardness ($n$)" in a less structured environment (e.g., Atari). Is there a proxy metric for $n$ that would allow one to apply the $\psi$ scaling law to a new domain? If not, clarifying that this model is currently descriptive rather than predictive for new environments would be helpful.

**4. Clarification of "Dispersion" in Table 2 (Minor):**
Table 2 lists "Dispersion" values of 4.1 and 8.1, noting that values $>1$ indicate the model is oversimplified. Please add a brief sentence interpreting the practical magnitude of these values. Is 8.1 considered a "good" fit in this context, or does it imply significant unmodeled dynamics? (The text suggests the latter, but more context for readers unfamiliar with this specific statistic would be good).

**5. Visual Clarity (Minor):**
In Figure 3 (Bottom row), the overlapping points can make it slightly hard to see the density of the "raw experimental data". Consider using a smaller marker size or slightly lower alpha transparency if possible to better visualize the distribution of successes vs. failures.

---

> ### Author Response · Authors · 2026-01-07
>
> We thank the reviewer for their detailed assessment! We strongly appreciate the positive feedback regarding the technical rigor of our paper and the validation of our "split batch" gradient moments derivation. We have addressed the requested changes as follows:
>
> **1. Environment Specificity and Transferability:** We agree that transferability to stochastic domains is a vital topic. We have expanded the Discussion section to explicitly analyze these limitations. We explain that while we chose DeepSea to isolate exploration hardness, extending this to stochastic domains is non-trivial due to the interplay of noise and reward density. We have added a paragraph to "Future Work" outlining the need for transferability.
>
> **2. Elaboration on "Cooperative Effects":** We agree that the shared replay buffer is the likely mechanism for the performance boost at small ensemble sizes ($K < 10$). We have expanded Section 4.2 to elaborate on this hypothesis. We discuss how the shared buffer likely induces cooperation by allowing agents to train on successful transitions found by others. We note that verifying this experimentally (e.g., by separating buffers) is difficult without breaking the fundamental ensemble architecture, so we have kept this discussion qualitative as requested.
>
> **3. Generalization of "Hardness" and Proxy Metrics:** We have added a discussion on potential proxy metrics (e.g. an effective horizon) that could allow practitioners to estimate hardness in unstructured environments. regarding the "descriptive vs. prescriptive" distinction: we have clarified in the text that while the model is currently descriptive for DeepSea, it provides a framework that could become prescriptive if a reliable metric for relative environment hardness is established.
>
> **4. Clarification of "Dispersion" in Table 2:** We have added a sentence interpreting the practical magnitude of the dispersion values. We clarified that a value of 8.1 implies significant unmodeled dynamics compared to a statistically ideal fit (normal residuals), which aligns with our analysis of the "U-shaped" residuals.
>
> **5. Visual Clarity in Figure 3:** We decreased the marker size and lowered alpha to make the density of the data easier to see.

---

### Review · Reviewer_Xp7H · 2026-01-05

**Summary Of Contributions:**

This paper presents a large-scale empirical study of Bootstrapped DQN (BDQN) and Randomized Prior BDQN (RP-BDQN) to characterize how exploration performance scales with ensemble size and task difficulty. The authors utilize the "DeepSea" environment, which allows for precise parameterization of exploration hardness.
1. The authors propose a scaling model, a simple "best-of-K" probabilistic model that predicts the probability of reward discovery.
2. Extensive experiments show RP-BDQN outperforms BDQN
3. The paper provides evidence that RP-BDQN's advantage stems from sustained ensemble diversity (measured via Q-value variance), whereas BDQN suffers from posterior collapse in non-convergent runs.

**Audience:**

Yes

**Audience Explanation:**

The audience include Deep RL Researchers (e.g., posterior collapse, setting ensemble K)

**Broader Impact Concerns:**

No concerns.

**Claims And Evidence:**

Yes

**Claims Explanation:**

*   **The Scaling Law:** The fit of the proposed model to the empirical data (Figure 3) supports this. The separation of the data into an "efficiency frontier" provides a visualization of the trade-offs. The authors also report the goodness-of-fit metrics ($R^2$ and dispersion), clarifying that the model is a first-order approximation rather than a perfect description.
*   **RP-BDQN Superiority & Diversity:** Figure 5 provides the distinction in diversity profiles between convergent and non-convergent runs. This supports posterior collapse is the failure mode for BDQN, and that Randomized Priors mitigate this.
*   The authors mentions the "independent trials" model works well. They support this not only by the model fit but also by analyzing where it fails (Figure 4). The analysis of the residuals (overestimating failure at low $K$, underestimating at high $K$) demonstrates a nuanced understanding of the data, rather than blindly fitting a curve.

**Requested Changes:**

The paper is well-written, technically sound, and offers clear insights. I have a few suggestions to strengthen the work, but none are critical for acceptance.

1. The study is exclusively conducted on DeepSea. How do you predict the proposed approach and scaling insights would transfer to other environments, like Atari Games (e.g., Montezuma's Revenge) or continuous control tasks, where the "hardness" cannot be easily quantified by a single parameter? Is there a proxy metric for n that practitioners could use to apply your scaling laws to these domains?
2. While the paper discusses ensemble size $K$, it would be beneficial to explicitly state the computational cost. Presumably, $K=10$ requires $10\times$ the FLOPs (or close to it). A brief mention of the "Performance vs. Compute" trade-off (not just Performance vs. K) would make the practical guidance even stronger.
3.  In the discussion, it might be worth explicitly mentioning that verifying this scaling law on a different family of environments (even a simple continuous domain) would be the logical next step to see if the "independent best-of-K" hypothesis holds outside of grid worlds.

---

> ### Author Response · Authors · 2026-01-07
>
> Thank you for a positive review!
>
> You might find it interesting that we have considered most of your points internally before, but, surprisingly, managed to fail to mention them in the manuscript.
>
> Proxy metrics: We have included a small paragraph in the discussion (5.1) about construction of hardness proxies. However, we are pessimistic about being able to identify such a proxy in general.
>
> FLOPS and K: if you look through Appendix B, even though it was never stated explicitly, you can tell that FLOPs are indeed proportional to K. Surprisingly, we never addressed this point in the main body. The point you have raised with regard to the efficiency trade-off is something we have failed to consider; therefore, we have added a plot next to residuals with the 'effective discovery rate', which quantifies the discovery rate per unit of compute (one greedy DQN run). The new content is located in 4.3, and left subplot of Figure 4.
>
> Finally, regarding the transfer to continuous domains, we have added a few sentences to the discussion outlining this as the logical next step.

---

### Decision · Action_Editor_FfWy · 2026-04-09

**Recommendation:** Reject

**Additional Comments:**

I recommend that the authors perform empirical analyses in at least one more environment, and then resubmit this paper.

**Audience:**

Yes

**Audience Explanation:**

The experiment results in Section 4 are interesting for researchers who have been working on exploration problems in reinforcement learning and who are familiar with the DeepSea environment.

All reviewers have answered "Yes" to this question and I agree with them.

**Claims And Evidence:**

No

**Claims Explanation:**

This paper has not proposed new algorithms or performed any mathematical analysis. It has also not proposed a new evaluation benchmark (the DeepSea environment was proposed in previous literature). The main contributions of this paper are the empirical analyses in Section 4, as well as the discussions in Section 5.

However, as Reviewer wnpu has pointed out, the empirical analyses of this paper are limited to one environment: DeepSea. I fully agree with Reviewer wnpu that to ensure the results of this paper convincing, the authors should perform empirical analyses in **at least one more environment.**

**Resubmission Of Major Revision:**

The authors may consider submitting a major revision at a later time.